# SkipOOD: Efficient Out-of-Distribution Input Detection using Skipping Mechanism

## Abstract

Detection of out-of-distribution (OOD) inputs has been a popular research direction in deep neural network (DNN) research, as OOD inputs can cause an undesirable decrease in the accuracy of the model. Furthermore, because of the deployment of DNNs on resource-constrained devices, it has become important to detect OOD inputs early during inference to save on computation. In this work, we focus on detecting OOD inputs in a partial inference setting and investigate whether the skipping mechanism used in dynamic neural networks (DyNNs) can be leveraged for early OOD detection. We first establish that the feature maps at various DyNN gates can help identify anomalies. Building on this, we propose SkipOOD, a lightweight OOD detector that uses an uncertainty scoring function and an exit detector at each gate to robustly identify OODs as early as possible. Through extensive evaluation, we demonstrate that SkipOOD achieves competitive performance in detecting OOD samples while reducing resource usage by nearly 50%.

## 1 Introduction

Large neural models (Wu et al., 2023) are frequently employed in safety-critical applications such as autonomous driving and security systems. Rejecting inputs the model is not designed to handle is critical for mitigating model safety concerns. In particular, detecting out-of-distribution (OOD) inputs (Liu et al., 2020), which are data points that significantly deviate from the training data distribution, can enhance the robustness of the model. OOD can refer to the distribution of features, labels, or both. For example, the pixels comprising a house number taken from Google Maps street view are OOD for a model that was trained to recognize handwritten zip codes despite the set of possible digits being the same for both domains. Conversely, when a model trained to recognize different breeds of dogs is presented with an image of a cat, the correct label falls outside of the training distribution. Both OOD scenarios can lead to incorrect predictions, which can incur a high cost for downstream applications, especially safety-critical ones, such as autonomous driving. Hence, it is very important to detect the OOD inputs to enhance model safety.

Additionally, it is not sufficient to merely detect the OOD inputs – they must be detected as early as possible without traversing the entire network layers. Early detection minimizes the usage of valuable computational resources and also helps reduce latency. Early detection of OOD could be beneficial in time-sensitive applications such as streaming financial fraud detection, as well as in resource-constrained deployments, e.g., for on-device image processing. Therefore, it is paramount to detect OOD inputs with high accuracy and promptness.

Multiple approaches have been proposed for OOD detection, including Bayesian Models, Deep Ensembles, and Feature Space Density Estimation (Gal & Ghahramani, 2016; Malinin & Gales, 2018; Lakshminarayanan et al., 2017; Mukhoti et al., 2023). However, those approaches rely on full or even multiple inference passes for OOD detection. Even efficient OOD detection techniques require outputs of either the final or penultimate layer, resulting in a full inference pass. As modern models can comprise billions of parameters, using a full inference pass to detect OODs can be costly.

Recent works (Lin et al., 2021; Zhou et al., 2023) have noted that low-dimensional features can be effective in detecting OODs (Ndiour et al., 2020), particularly when using additional low-dimensional intermediate

outputs from dynamic neural networks (DyNNs)(Han et al., 2021). DyNNs, which selectively activate parts of the network based on intermediate outputs, offer a promising direction for reducing inference costs. While previous studies have explored early-exit DyNNs for OOD detection (Lin et al., 2021; Zhou et al., 2023), their success depends on addressing two main challenges: developing a control mechanism that balances computational cost and detection accuracy, and ensuring the control subnetwork adapts to variations in model size and architecture with minimal complexity overhead.

Following this motivation, in this work, we focus on OOD detection, leveraging the capabilities of a specific type of DyNNs called Conditional-Skipping DyNN (CS-DyNNs) (Wu et al., 2018; Wang et al., 2018), which uses gates to control the execution of every block. Our preliminary study in Section 2 shows that the gate features are low-dimensional and some of the gates have a strong capability to detect OODs. Based on that insight, we propose SkipOOD, a novel OOD detection technique based on CS-DyNNs. SkipOOD consists of two components: OOD Scoring Function and Exit Detector. The OOD scoring function is an efficient low-cost uncertainty measurement method that can detect whether an input is OOD or not. The Exit Detector uses a patience-based exit strategy to select which gate will be used as the exit if an input is detected as OOD.

We evaluate SkipOOD on two aspects: effectiveness and efficiency. Through measuring effectiveness, we evaluate the OOD detection capability of SkipOOD against baselines. Through efficiency measurement, we discuss the latency caused by SkipOOD for OOD detection. Based on the evaluation on three baselines and ten OOD datasets, we find that SkipOOD can detect OOD examples with 0.87 AUROC score reducing 50% of the latency.

Our work makes the following contributions:

- **Problem Formulation.** Our work is the first to use Conditional Skipping DyNNs to detect OOD examples efficiently.

- **Approach.** We propose SkipOOD, a novel OOD detection technique that consists of two components: One focusing on the OOD scoring function and the other focusing on exit detection.

- **Experimentation.** We investigate Conditional Skipping DyNNs's capability to detect OOD by using two models, two ID (in-distribution) datasets, and ten OOD datasets.

## 2 Background and Motivation

**OOD Detection.** The literature around OOD detection is vast, with surveys (Yang et al., 2021) and (Salehi et al., 2022) providing extensive reviews. Distribution shifts in the inputs can occur in practice for several reasons such as a novel class, an unusual context or deliberate adversarial perturbation. Some efforts such as (Bulusu et al., 2020; Geng et al., 2020; Karunanayake et al., 2024) explicitly account for the shift characteristics in their framework. In contrast, we treat all such shifts in an agnostic manner and propose a OOD detection mechanism in a unified setting. Various research directions have been proposed to enable deep-learning-based OOD detection and scoring based detection (Hendrycks & Gimpel, 2016; Mukhoti et al., 2021; Liu et al., 2020; Wang et al., 2021) is a dominant paradigm. For SkipOOD, we focus on two specific mechanisms that are also score-centric: feature density based (Mukhoti et al., 2021) and energy-based (Liu et al., 2020).

**Dynamic Neural Networks.** Dynamic Neural Networks (DyNNs) (Han et al., 2021) provide a computationally efficient inference mechanism by adaptively modifying their network structure according to the input data characteristics. In particular, Conditional-Skipping DyNNs (CS-DyNNs (Wu et al., 2018; Wang et al., 2018)) (see Figure 1) reduce network depth for certain input instances on-the-fly by selectively bypassing intermediate layers through the use of skip connections. In contrast to networks that employ early termination strategies (Dai et al., 2020), these CS-DyNNs offer fine-grained, plug-and-play control, allowing them to effectively handle inputs with varying degrees of complexity. Our work builds on top of CS-DyNNs.

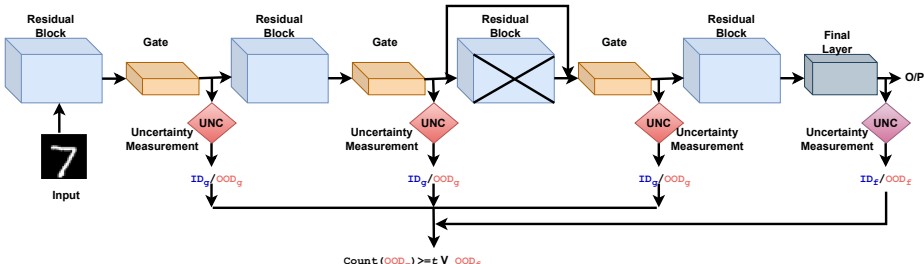

Figure 1: SkipOOD pipeline. Inputs are assessed for OOD at each gate using uncertainty scores. Inference is halted for an OOD example based on a voting mechanism that pools information across multiple gates.

Formally, let $x^l$ and $\mathcal{H}^l$ denote the input feature map and the network function at layer $l$ respectively. The output at layer $l+1$ is computed for CS-DyNNs as

$$x^{l+1} = \mathbb{1}(\phi_g(x^l) > \epsilon^l)\mathcal{H}^l(x^l) + x^l, \tag{1}$$

where $\phi_g : x \rightarrow [0,1]$ is a gating function that determines whether a layer must be executed, and $\mathbb{1}$ is the indicator function. In practice, the gating functions in CS-DyNNs are not ubiquitously introduced after every layer. Instead, they are strategically placed after specific blocks, such as subsequent to a residual block or a group of convolutional layers. In the above, $l$ is the layer immediately preceding a gate $g$. The threshold parameter $\epsilon$ provides a flexible mechanism to tune the trade-off between accuracy and efficiency.

**Motivation.** Can the gating mechanism in CS-DyNNs also detect out-of-distribution (OOD) inputs? To verify this, we train a SkipNet (Wang et al., 2018) model with CIFAR-10 (Krizhevsky et al., 2009) dataset, and observe the Deep Deterministic Uncertainty (DDU) (Mukhoti et al., 2023) scores of the feature map at the various gates for both in-distribution (ID) and OOD examples. Here CIFAR-10 is used for ID samples while SVHN (Netzer et al., 2011) is used for OOD samples. The DDU scores quantify the uncertainty in the predictions, and are useful in identifying outliers or anomalies. Figure 8 (in appendix) illustrates that OODs can indeed be detected at the gates. However, accurate detection is achieved only at a limited subset of gates (e.g. Gate 6, 18 and 29). Furthermore, even the gates located closer to the final output (e.g. Gate 35), which have the potential to access more comprehensive feature maps, exhibit inconsistent performance in identifying OOD samples.

This limitation motivates our pursuit of a solution that aggregates information from several CS-DyNNs gates in order to robustly detect OOD inputs. We also design a light-weight multi-score approach for OOD detection, that capitalizes on the complementary strengths of diverse uncertainty measures while being resource efficient.

## 3 Approach

Our proposed solution for efficient OOD detection in DyNNs is designed to seamlessly integrate with existing architectures. It consists of two novel components: (a) a scoring function that evaluates the likelihood of an input instance being OOD at each CS-DyNN gate, and (b) an exit detector that determines whether the input should be classified as OOD. These components can be easily incorporated into current DyNN models with minimal modifications.

Unlike conventional methods that compute OOD estimates solely at the final layer, our approach conducts OOD estimation at every gate, in addition to the final layer. This enables us to account for different levels of abstraction and noise in the data, combine the strengths of multiple layers, and, crucially, track and monitor the variations in uncertainty scores. Moreover, calculating the scores at each gate aligns with the specific intervals chosen by the CS-DyNN model designer and provides a more tailored OOD detection mechanism. Below, we discuss the two aforementioned components in detail, along with the SkipOOD pipeline.

Table 1: Comparison results for OOD detection efficiency of SkipOOD against baselines using the mean latency (Sec) aggregated over all the ten test datasets.

| Dataset | Model | SkipOOD(D,D) | SkipOOD(E,E) | SkipOOD(D,E) | SkipOOD(E,D) | MOOD | DDU | ES | ML | MSP | KLM |
|---|---|---|---|---|---|---|---|---|---|---|---|
| CIFAR10 | RN110 | 0.5 | 0.4 | 0.5 | 0.6 | 0.4 | 0.83 | 0.83 | 0.83 | 0.83 | 0.84 |
| | RN74 | 0.3 | 0.3 | 0.4 | 0.5 | 0.3 | 0.64 | 0.64 | 0.64 | 0.64 | 0.65 |
| CIFAR100 | RN110 | 0.5 | 0.5 | 0.7 | 0.7 | 0.4 | 0.83 | 0.83 | 0.83 | 0.83 | 0.84 |
| | RN74 | 0.3 | 0.3 | 0.5 | 0.5 | 0.3 | 0.64 | 0.64 | 0.64 | 0.64 | 0.65 |

## 3.1   OOD Scoring Functions

To offset the computational overhead introduced by OOD estimation at every gate, we employ a lightweight scoring mechanism to quantify the uncertainty of an input example in the context of a trained model. We eschew methods that require multiple forward passes during inference (Ovadia et al., 2019) and favor those that have a low footprint. Specifically, we consider two different scoring mechanisms to detect OOD: feature density-based and energy-based.

Recent studies (Mukhoti et al., 2023) have shown that feature space density estimators can reliably capture epistemic uncertainties, which often correspond to unseen examples during training, and hence are useful for OOD detection. The key idea here is to first estimate the probability density of the feature representations in the latent space of a trained model, and then calculate the likelihood of a given sample against this density function to determine whether it was previously seen. Although there are several choices for density estimation, we employ a Gaussian Mixture Model (GMM) with separate mean and covariance parameters for each possible output category of a gate. This technique does not require any OOD samples or offline training and can be evaluated in a single pass.

Formally, the density-based score $DDU$ at gate $g$ and the corresponding OOD categorization is written as

$$OOD_g(x) = \sum_{k=1}^{K} \mathcal{N}(\mathcal{H}^l(x^l); \mu_{g,k}, \sigma_{g,k}) < \lambda_g^d \tag{2}$$

where $K$ denotes the number of classes predicted by a gate (typically two) and $\mu$ and $\sigma$ are the mean and covariances of a Gaussian distribution, respectively. This density score is expected to be low for an OOD input while being high for an ID sample. In the above, $\lambda$ is a threshold parameter that can be configured for individual gates or optionally be reused across all the gates.

Besides the above probabilistic approach to quantify the OOD likelihood, we also employ an energy function-based method (Grathwohl et al., 2020) that measures the compatibility of an input sample with the learned model. Specifically, we compute an energy score from the unnormalized activation outputs of a gate in order to provide more discriminative information about an input sample. Let $\Phi_g$ be the pre-softmax output of a gating function $\phi_g$. The energy based score $ES$ at gate $g$ and its OOD category are computed as follows, with $\lambda$ being a threshold parameter.

$$OOD_g(x) = \log \sum_{k=1}^{K} \exp(\Phi_{g,k}(x^l)) < \lambda_g^e \tag{3}$$

The unnormalized logits is likely to have a small value for unseen samples, and consequently their energy scores will be lower when compared with ID inputs.

We conduct extensive evaluations to compare the effectiveness of both these scores in the context of CS-DyNNs. Although we could use a weighted combination of these scores, their distinct ranges would make it challenging to normalize them automatically and leave it for future work.

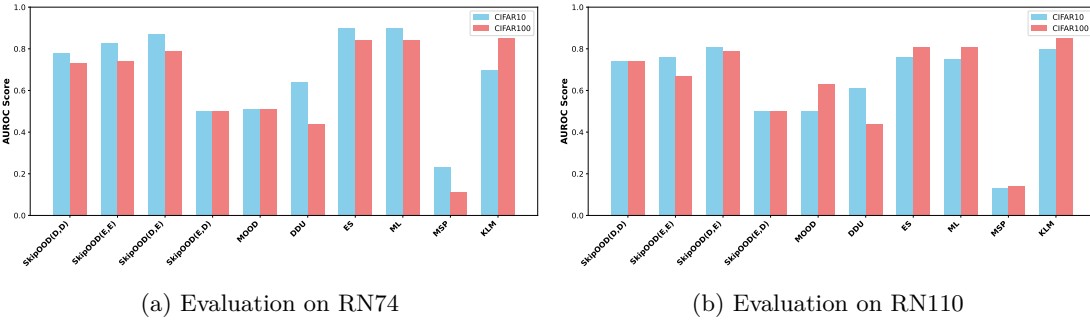

(a) Evaluation on RN74  (b) Evaluation on RN110

Figure 2: Comparison results for OOD detection effectiveness using the mean AUROC scores aggregated over all the ten test datasets.

## 3.2   OOD Exit Detector

Timely truncation of network computation upon detecting OOD inputs is crucial for resource efficiency. However, naively exiting the network as soon as a gate classifies an input as OOD can be unreliable, especially with inputs of varying complexity or errors in score computation. Despite attempts to model input complexity (Lin et al., 2021), detecting OOD based on a single gate remains problematic due to the intricate nature of feature spaces in DyNNs.

We propose an exit strategy that aggregates information from multiple gates to classify inputs as OOD. Using a cumulative sum technique, we pool OOD results across several gates, and if a threshold number of gates identify the input as OOD, we conclude the input is OOD and exit the network. Unlike (Zhou et al., 2020), our approach does not require consecutive gate agreement, instead employing a majority voting method that accommodates noise and errors. Let $t$ be a threshold denoting the minimum number of gates that must agree, and $G$ be the total number of gates. The decision to exit at a gate $g \in \{1, ..., G\}$ is computed as $Exit(g, x) = \left[ \sum_{g'=1}^{g} OOD_{g'}(x) \right] \geq t$.

The aforementioned criterion dynamically halts the inference process only when a sufficient number of detectors concur that the input is an OOD sample. This approach effectively balances the trade-off between accuracy and efficiency, ensuring robust performance while minimizing computational overhead.

## 3.3   SkipOOD Pipeline

Figure 1 shows the input processing pipeline of SkipOOD. For an input provided to our model, uncertainty scores at a given gate are calculated using either feature density or energy based techniques, and subsequently the input is categorized as OOD or ID. If $t$ different gates agree that the input is OOD, the inference is stopped to minimize computations. If the consensus is not reached, the computation proceeds to the next gate, repeating the above procedure until the final layer. Inputs that successfully traverse the entire network are considered ID inputs.

## 4   Evaluation

We assess SkipOOD on two key aspects: its effectiveness in accurately detecting OOD inputs and its efficiency in minimizing resource usage. We discuss below the evaluation setup and results. See supplementary material for more in-depth discussions.

## 4.1   Setup

**Datasets and Architectures.** We consider CIFAR-10 and CIFAR-100 datasets (Krizhevsky et al., 2009) as ID data. We consider ten different datasets for OOD data. These datasets are MNIST (LeCun et al., 1998), K-MNIST (Clanuwat et al., 2018), Fashion-MNIST (Xiao et al., 2017), SVHN (Netzer et al., 2011), LSUN

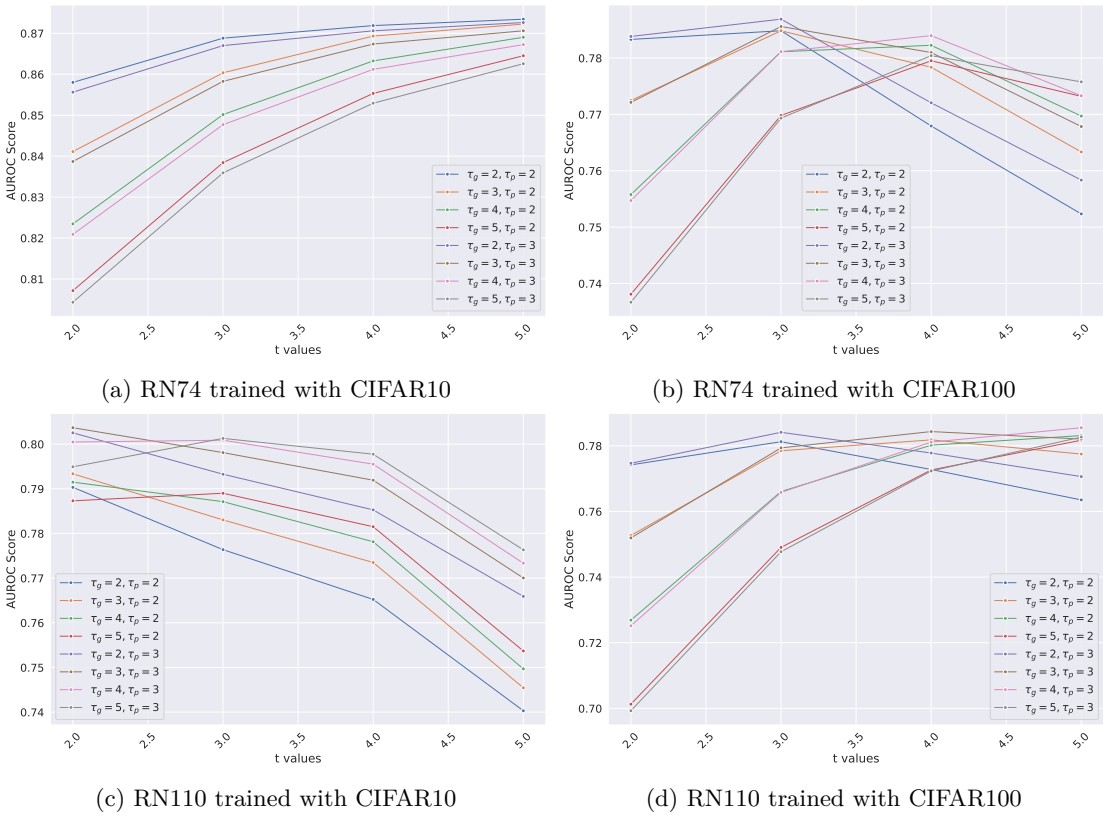

Figure 3: Impact of hyperparameters $t$ and $\tau_g$ on effeciveness of SkipOOD. $\tau_p = 2$ and $\tau_p = 3$ were used.

(Crop) (Yu et al., 2015), LSUN (Resize) (Yu et al., 2015), Textures (Cimpoi et al., 2014), STL10 (Coates et al., 2011), Places-365 (Zhou et al., 2017), iSUN (Xu et al., 2015).

We use ResNet (He et al., 2016) as the backbone architecture for DyNNs, particularly ResNet-74 (RN74) and Resnet-110 (RN110). We train SkipNet (Wang et al., 2018) DyNNs for evaluating our model, while we train EENet (Demir, 2019) DyNN for evaluating the MOOD baseline. ResNet-74 has 36 gates, while ResNet-110 has 54 gates.

**Baselines.** We compare our method against MOOD (Lin et al., 2021), a state-of-the-art approach for detecting OOD in DyNNs. Additionally, we use five additional static OOD detection techniques which use final layer outputs to detect OODs: energy scores (ES) (Liu et al., 2020), DDU scores (Mukhoti et al., 2023), KL Matching (KLM) (Hendrycks et al., 2019), Maximum Softmax Probability (MSP) (Hendrycks & Gimpel, 2016), MaxLogit (ML) (Hendrycks et al., 2019).

**Settings.** We consider OOD detection at two different levels of abstraction: at the intermediate gates and at the final layer. The former is essential for early detection, while the latter is for the ability to process richer feature representations. When including the different uncertainty scores discussed in section 3.1, this provides us four evaluation combinations: (I) **SkipOOD(D,D).** Density score-based OOD detection at both gate level and at the final layer. (II) **SkipOOD(E,E).** Energy score-based OOD detection, at both gate level and at the final layer. (III) **SkipOOD(D,E).** Density score-based OOD detection at the gate level and energy score-based at the final layer. (IV) **SkipOOD(E,D).** Energy score-based OOD detection at the gate level and density score based at the final layer.

**Hyperparameters.** We consider three different hyperparameter configurations during evaluation: percentile threshold of the in-distribution range of uncertainty scores at various gates ($\tau_g$), percentile threshold of the in-distribution range of uncertainty scores at the final layer ($\tau_p$), and the number of gates needed to discard an OOD input ($t$). The $\tau$ values are calculated by first determining the range of scores in the

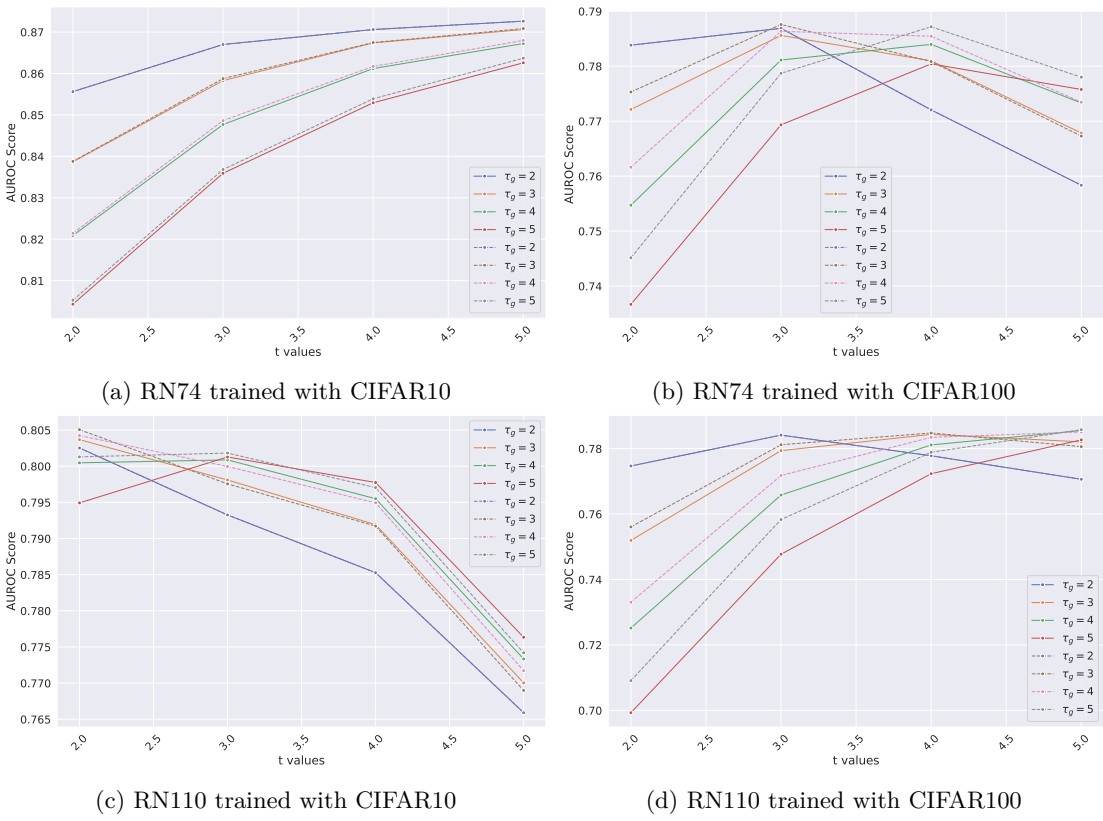

Figure 4: Impact of a stricter threshold on $\tau_g$ for OOD detection effectiveness. Dotted lines represent the strict threshold setting.

ID training inputs and then applying a percentile value to that range. Specifically, we use the second, third, fourth, and fifth percentiles. We use values 2, 3, 4, and 5 for $t$. See Table 2(in appendix) for the configurations used for the various backbones and datasets.

## 4.2 Effectiveness

Figure 2 compares the performance (mean AUROC score) of our proposed solution against various baselines in detecting OOD inputs accurately. See Figures 9 and 10 in appendix for a detailed analysis. In general, we observe that the SkipOOD variants outperform baselines like MOOD, DDU, MSP by a considerable margin across different configurations. Additionally, SkipOOD's effectiveness in detecting OOD's is in the same range of MaxLogit, Energy Score and KL Matching techniques. Note that the other methods utilize the entire network, disregarding any resource efficiency criteria. Despite accounting for computational efficiency using early exits, our SkipOOD solution surpasses or matches the performance of all the popular baselines. The underwhelming results of the DDU and MSP scores computed solely at the final layer also highlight the importance of including a robust detection mechanism that spans different levels of feature representations, which our method proposes.

When using mixed combinations for the gate level and final layer, we observe that SkipOOD(E,D) yields low AUROC scores. This may be due to the sparsity introduced by the binary gate classifier and the challenges density-based mechanisms face in handling the different categories present in the final layer. Additionally, the MOOD baseline struggles in most configurations, underscoring its limitations with complex inputs. Overall, SkipOOD(D,E) performs best, effectively combining density-based scores for varying abstraction levels with energy score methods at the final layer.

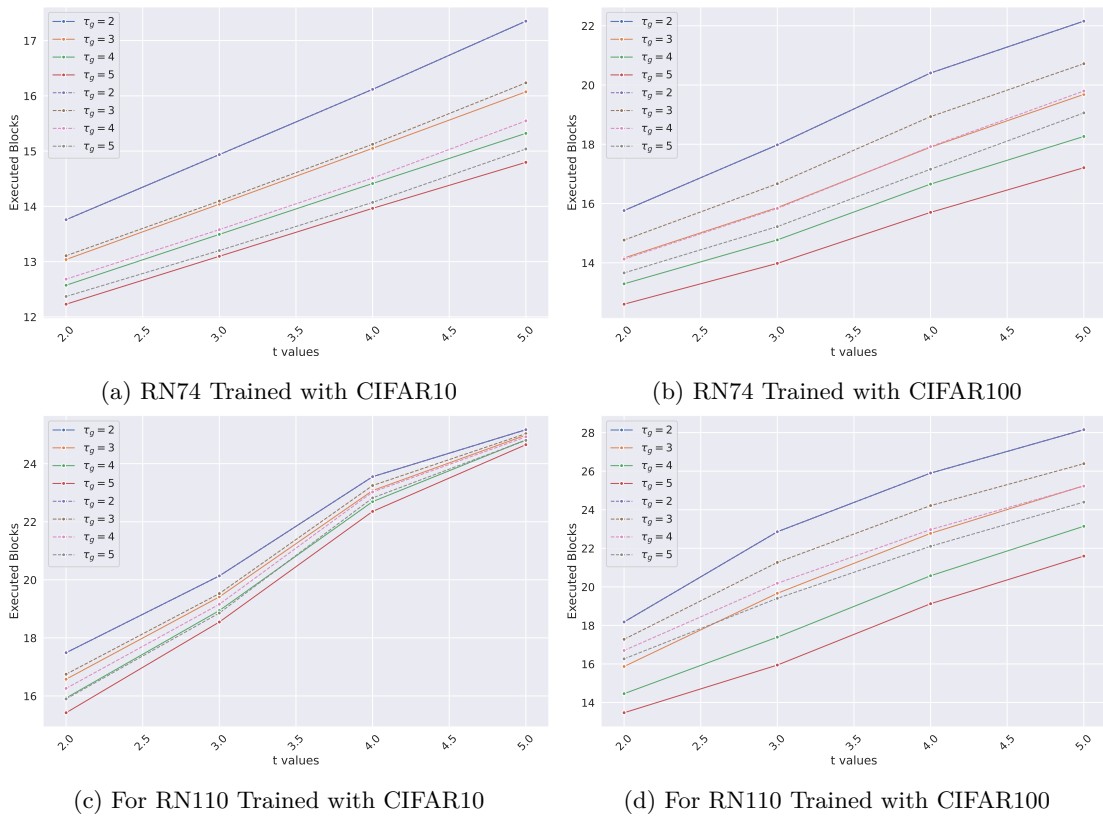

(a) RN74 Trained with CIFAR10        (b) RN74 Trained with CIFAR100

(c) For RN110 Trained with CIFAR10      (d) For RN110 Trained with CIFAR100

Figure 5: Impact of a stricter threshold on $\tau_g$ for computational efficiency. Dotted lines represent strict threshold setting.

## 4.3 Efficiency

Does SkipOOD reduce the computation cycles for OOD inputs? We show the mean latency caused by the different methods in Table 1 (see Table 3 in the appendix for detailed analysis) and note that our method consumes significantly less time for OOD detection than the static baselines, which traverse the entire network. The mean latency caused by KL-Matching is slightly higher than other static baselines, which is because of the additional computations caused by the KL-Matching algorithm. We also find that time consumed by SkipOOD(D,E) to detect OODs is in the same range as MOOD's time consumption. It also appears that using only DDU or ES would also cause comparably higher latency than SkipOOD(D,E). To summarize, the SkipOOD(D,E) method provides the optimal balance between accurately detecting OODs and minimizing the the network latency.

# 5 Additional Insights

Here, we discuss additional experimental insights.

## 5.1 Parameter Sensitivity

In this section, we discuss the impact of hyper-parameters like $\tau_g$ and $t$ on the OOD detection capability of SkipOOD. We consider two values of $\tau_p$ (2nd and 3rd percentile) and plot the mean AUROC scores aggregated over ten datasets for different values of $\tau_g$ and $t$. We employ SkipOOD(D,E) for this exercise since it is the best-performing model.

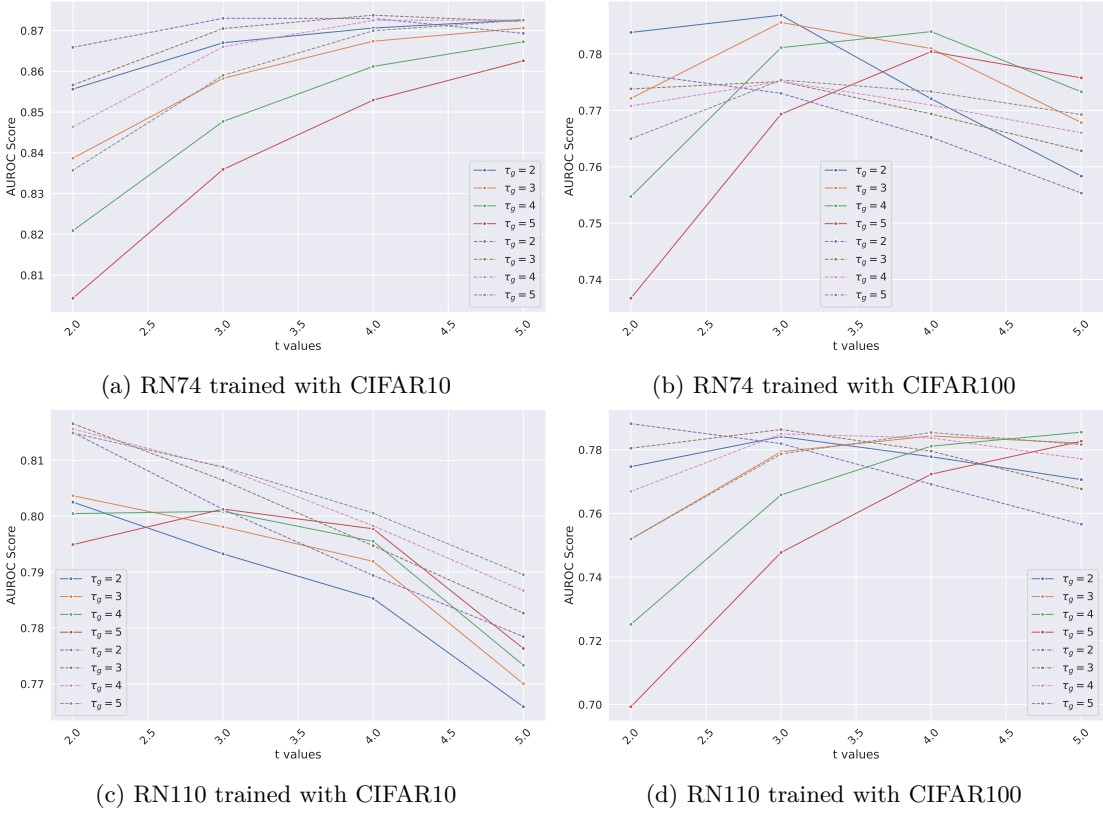

Figure 6: Impact of accumulating gate features for OOD detection effectiveness. Dotted lines represent the accumulated features setting.

Figure 3 shows these plots, and it can be seen that for all the combinations, the relative percentage difference between the highest and lowest mean AUROC values is well within 10%. This observation suggests that changing the hyper-parameter values does not have a drastic effect on the AUROC scores, thereby highlighting the stability of the model. However, it may be prudent to tune these parameters when dealing with a large dataset that has a high volume of OOD samples.

Additionally, for higher values of $t$, it is noticeable that the average AUROC scores improve for three out of four cases (Figure 3a, 3b, 3d). For small values of $t$, fewer gates decide if an input is OOD or not. This might result in a significant number of ID examples being misclassified as OOD and end up reducing the AUROC score. The observation is not followed for Figure 3c. The difference between the highest and lowest mean AUROC is within 1% relative difference, showing that having a low value for $t$ might not be beneficial.

## 5.2 Impact of Stricter Threshold

We discuss here the impact of the threshold parameter $\tau_g$ on SkipOOD's performance. We employed a constant threshold across all the gates in our experiments. In practice, the capability and functionality of each gate might differ and it would be useful to examine whether altering these thresholds can be beneficial. For example, the gates at the early part of the network often mis-detect OODs since they would not have access to rich feature representations. Hence, we study the effect of stricter $\tau_g$ thresholds at the early gates. To be precise, if a gate number is less than one-sixth of the total number of gates, we consider the gate to be an early gate. We had used $\tau_g$ values of 2, 3, 4, and 5 in our evaluation, but here we set $\tau_g = 2$ for all early gates and use the same values as before for the rest of the gates. For fair comparison, we freeze $\tau_p$ value to be 3. Here as well, we consider only SkipOOD(D,E).

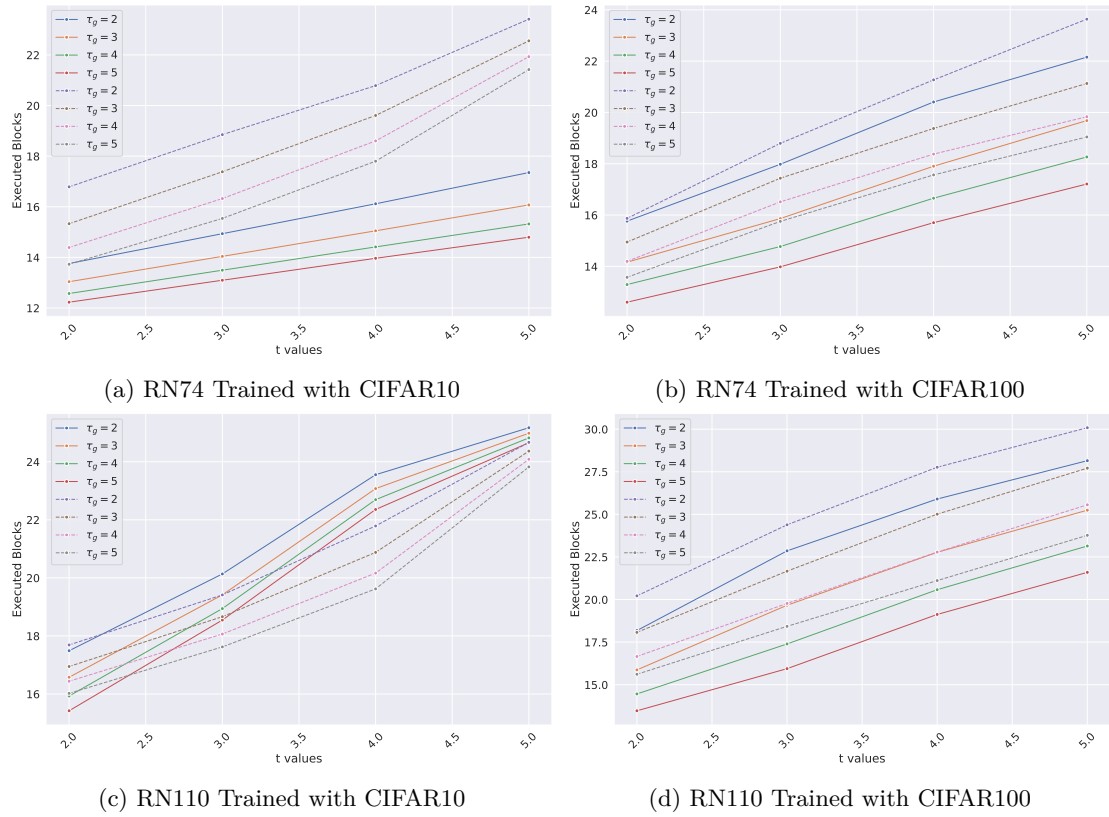

(a) RN74 Trained with CIFAR10          (b) RN74 Trained with CIFAR100

(c) RN110 Trained with CIFAR10          (d) RN110 Trained with CIFAR100

Figure 7: Impact of accumulating gate features for computational efficiency. Dotted lines represent strict accumulated features setting.

Figure 4 shows the impact of this stricter threshold on the SkipOOD effectiveness. The solid lines represent the evaluation with a constant $\tau_g$ across the model, while the dotted lines represent the evaluation with stricter $\tau_g$ values at the early gates. It can be observed that for a majority of the configurations, a strict threshold for early gates produces better AUROC scores. This is due to the fact that fewer ID examples are mis-predicted as OOD in earlier gates, thereby improving the AUROC scores.

However, a strict threshold for early gates may result in reducing the resource efficiency of the model. Figure 5 confirms this hypothesis. We report the mean number of blocks required to detect OOD, and it can be seen that the efficiency of SkipOOD indeed decreases if we set strict thresholds of $\tau_g$ for early gates. This trade-off can be exploited to seek an optimal balance between OOD detection effectiveness and computational efficiency, depending on the target application.

## 5.3 Feature Accumulation through Multiple Gates

In this section, we discuss if feature accumulation through multiple gates could improve the OOD detection capability of the SkipOOD or not. For that purpose, we create $N/2$ exits out of $N$ gates, where each exit would accumulate the features of two gates and measure uncertainty based on accumulated features.

Figure 6 shows the AUROC scores of SkipOOD(D,E) with (dotted lines) and without accumulated features setting. It is noticeable that for three out of four model-dataset pairs, the maximum AUROC score is higher for the setting with accumulated features. Also, with the accumulated features setting, we find that the AUROC scores are higher for lower $t$ values. However, we find the similar pattern of using more number of blocks for OOD detection ( three out of four model-dataset pairs) while using accumulated features setting. The results can be found in Figure 7.

## 6 Conclusion

We propose in this paper SkipOOD, a robust and resource-efficient OOD detection technique based on the skipping mechanism of dynamic neural networks, aimed at developing safe AI systems. We introduced two different OOD scoring functions that leverage the simplicity of density based methods and the discriminative power of energy functions. We also proposed a novel exit detector that aggregates information from several gates to identify termination criteria. Our comprehensive evaluation on ten different datasets, assessing both effectiveness and efficiency, illustrate the utility of our solution. Additional insights are provided in the appendix.

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
