# OpenReview forum: "SkipOOD: Efficient Out-of-Distribution Input Detection using Skipping Mechanism"
_TMLR — Rejected by TMLR_

### Review · Reviewer_a7he · 2024-12-13

**Summary Of Contributions:**

The paper introduces SkipOOD, a method for efficiently detecting out-of-distribution (OOD) inputs by leveraging the skipping mechanisms inherent in Conditional-Skipping Dynamic Neural Networks (CS-DyNNs). Traditional OOD detection methods often require full inference passes, which can be computationally intensive. In contrast, SkipOOD utilizes intermediate features from CS-DyNN gates to identify OOD inputs early in the inference process, thereby reducing computational overhead.

The authors' preliminary analysis indicates that certain gate features within CS-DyNNs are particularly effective at detecting anomalies. Building on this insight, SkipOOD employs an uncertainty scoring function alongside an exit detector at each gate to promptly and accurately identify OOD inputs.

This approach is potentially useful for applications in resource-constrained environments, such as on-device image processing, where both computational efficiency and rapid OOD detection are critical.

**Audience:**

No

**Claims And Evidence:**

Yes

**Requested Changes:**

I suggest moving the detailed numbers on each OOD datasets into the main paper.
Also I suggest converting the bar charts for large set of numbers into the tables for better visual quality.

**Strengths And Weaknesses:**

Strength:

The paper leverages the inherent gating mechanisms of Conditional-Skipping Dynamic Neural Networks (CS-DyNNs) for OOD detection. This novel integration allows SkipOOD to identify OOD inputs early in the inference process, which will be useful in particular domains.

SkipOOD achieves a reduction in computational overhead, with resource usage reduced by nearly half compared to traditional OOD detection methods.

Weakness:

Challenge in Handling Texturally Similar but Semantically Different OOD Samples: While early exiting at intermediate layers provides efficiency benefits, it introduces challenges in detecting OOD samples that are texturally similar but semantically different. For instance, images of simple stick drawings might evade detection if the model relies heavily on early-layer features that are insufficient for capturing semantic differences. This limitation undermines the robustness of SkipOOD in scenarios where fine-grained semantic distinctions are critical.

Limited Contribution: The methodological contribution of SkipOOD appears incremental than prior works, such as Lin et al. (2021), which have already explored early-exiting strategies for efficiency gains. While SkipOOD utilizes another gating mechanism within CS-DyNNs, this approach represents an adaptation rather than a substantial conceptual advancement. The lack of a clear differentiation from existing methods reduces its impact on the research community.

Insufficient Experimental Benchmarking:
The experimental evaluation lacks comprehensiveness, particularly in the context of recently developed benchmarks. For instance, the authors have not included comparisons with state-of-the-art methods on the OpenOODv1.5 benchmark. Adding evaluations against these newer methods would strengthen the empirical evidence and position SkipOOD more effectively.

---

> ### Author Response · Authors · 2025-01-09
> **Response to Reviewer a7he**
>
> We thank the reviewer for their comments.
>
> **OpenOOD V1.5 Benchmark**:
>
> In our current experiments, we evaluate our approach on far OOD datasets as part of the OpenOOD V1.5 benchmark, specifically using C10 and C100. In the revised version, we plan to include evaluations on near OOD datasets as well.
>
> **Handling Texturally Similar Data**:
>
> SkipOOD's termination decision is based on the uncertainty in estimating whether an input is OOD, allowing it to incorporate outputs from later-stage layers depending on input characteristics. We acknowledge the potential risk of premature termination for texturally similar but semantically different OOD samples. However, this risk can be mitigated through proper calibration, which balances accuracy with resource efficiency.
>
> **Incremental Contribution**:
>
> In previous studies, CS-DyNN gates were solely utilized to decide whether to skip a layer. In contrast, we introduce a novel, multi-purpose application of these gates, extending their use to include OOD detection and early termination—areas that have not been previously explored. Additionally, the exit strategy we propose is more robust than those in existing works, such as MOOD.

---

### Review · Reviewer_sbH5 · 2024-12-19

**Summary Of Contributions:**

The authors adopt a dynamic neural network baseline called SkipNet (Wang et al., 2018) and adapt it for supervised OOD detection on small-scale benchmarks like CIFAR10 → SVHN. They present two different adaptations of using information from the gates to derive their OOD score called SkipOOD, one using the existing energy score of a binary gate and the Deep Deterministic Uncertainty score to estimate the likelihood of a given test sample (fitting a Gaussian). In addition, a hand-crafted mechanism is proposed for early OOD detection (OOD Exit Detector, Sec 3.2), meaning to avoid performing a full forward pass. The idea is to derive consensus from the outputs of the gates given a threshold t (i.e. number of gates detecting OOD samples) without the need to process the image. Consistent results from 2-4 gates that the input is OOD. Such an application might be beneficial for edge devices with limited power.

**Audience:**

No

**Claims And Evidence:**

Yes

**Requested Changes:**

I have provided very specific comments that the author could address to improve the quality and readability of their manuscript.

**Strengths And Weaknesses:**

### Strengths: The idea of applying OOD detection scores in the binary “gates” of dynamic neural networks has not been previously explored.

# Major concerns:

## 1. Baselines and benchmarks.

I find the overall presentation of the paper quite outdated in terms of methods (SkipNet 2018), models (ResNets 2016), and benchmarks (OOD detection using CIFAR10 and SVHN as out-distribution). It’s been more than 2 years since the evaluation has moved to large-scale datasets such as ImageNet, and the current literature is mainly focused on unsupervised approaches, meaning when in-distribution labels are not available. Specifically, after Fort et al. [3], supervised OOD detection on small-scale datasets is considered “solved”, meaning that small-scale benchmarks are saturated.

While I am not an expert in the field of Dynamic Neural Networks (DNN), it takes a few minutes to find newer approaches (from ICLR 2022 [1]) that develop DNN with publicly available weights (see ref) on Imagenet [1]. Adopting competitive baselines would add credibility to this work, as older methods are easier to improve in hindsight, i.e. after 5 years of research advancements. There might be even newer or more suitable approaches. Using publicly available weights would also make the results more reliable due to easier reproducibility, which is something that is highly valued in TMLR, as far as I understand.

Finally, these OOD detection scores, like the energy score, exist and have been applied to the logits of a classifier. Applying them to the output of a binary gate is not especially novel. However, this approach can only be applied to specific neural networks (dynamic), which are not been actively used in recent years, to the best of my knowledge. I don’t think there are many researchers in the TMLR audience of 2025 that would be interested in something so niche, while the results seem questionable to me (see below).

Future suggestion: one fair and strong baseline would be to somehow compare the earlier features of the supervised encoder (i.e. early output convolutional block of a resnet) with the proposed early-stage detection. One example may be the scores from Park et al. [4].

## 2. Table 1.

Table 1 is presented on page 4, but there is no reference whatsoever until page 8. “Efficiency” is achieving the same or similar performance (i.e. AUROC) with less compute, while Table 1 only reports the inference time, to the best of my understanding.

What are the authors referring to by “ten test datasets”? I thought SVHN was the test OOD dataset, as the authors write:

“Here CIFAR-10 is used for ID samples while SVHN (Netzer et al., 2011) is used for OOD samples.” (page 3).

This is quite confusing to me. I had to read the experimental part to see the ten datasets mentioned in the paper.

The baseline scores, such as the energy score (ES), are presented on page 6. How is the reader supposed to know what do the authors mean by ES in Table 1? The same applies for all other abbreviations. Additionally, all baseline post-hoc OOD detection scores (i.e. MSP, ES) that perform a forward pass are expected to have similar inference times, so the information provided is not that useful.

## 3. Figure 2.
I don’t believe the reported AUROCs for post-hoc methods. Maybe the authors have made a mistake with the scale of the y-axis. According to the reported numbers, the maximum softmax probability (MSP) of a supervised ResNet74 and a ResNet101 is less than 50%, meaning that the model performs worse than a random guess on CIFAR10→ SVHN and CIFAR100 → SVHN, some of the easiest benchmarks on the field of visual OOD detection.

As an example, I have trained a small resnet18 myself on cifar10 using the well-studied state-of-the-art hyperparameters and configuration from [2], and I have obtained the following AUROC score using a standard MSP evaluation:
```
CIFAR10 --> SVHN , MSP AUROC: 94.43
CIFAR10 --> CIFAR100, MSP AUROC: 86.80
```
I can provide the code and checkpoint if the authors or other reviewers are interested. In conclusion, I don’t find the reported results particularly competitive, while there is an additional complexity by considering variations of gated convnets.

Can the authors take a state-of-the-art supervised resnet18 or resnet50 on cifar10 and cifar100, add the gating mechanism, and get better supervised OOD results? That would at least prove (experimentally) that there is some benefit to the added complexity of going through such a specific architecture.



## 4. Undefined abbreviations of OOD scores.
While the paper focused on OOD detection, the OOD score abbreviations are not sufficiently explained, making the manuscript hard to read. While the MSP is quite known as an abbreviation, the energy score and others are not that established.

## 5. Motivation:
“Can the gating mechanism in CS-DyNNs also detect out-of-distribution (OOD) inputs?” I think the motivation statement should have been more broad. For example, can dynamic neural networks match the OOD detection performance of standard supervised models? If yes, can they achieve similar OOD detection performance as measured by AUROC significantly faster? This is related to concern number 3 above.


## 6. OOD exit detector

The equation “Exit(g, x)” (Sec 3.2) is likely not properly formulated. I assume you mean that given an OOD detection threshold (called lambda in this work) needs to be fixed to decide whether a particular gate identifies the input as OOD. In other words, the gate makes a “hard” decision 0 or 1. Then, one needs to fix the second hyperparameter, t, an integer that is the maximum cumulative sum. If I assume correctly, t is an integer, which means the line plots do not make any sense. A scatter plot is ok, though (fig 3 and 4).

The three tau values indicated with different subscripts look quite confusing to me the way it is written, and their choice looks quite “hacky” at first glance. How would these decisions transfer to another in-distribution or another network architecture?


## 7. Cryptic method names SkipOOD(X,Y)  and questionable usefulness.

The abbreviation SkipOOD(D,E) is not explained very well in my humble opinion. Additionally, the proposed method always includes the output of the final layer which means that every time an image is processed until the end, there is no speedup compared to a standard score based on the classifier’s logits like MSP and energy. It is not clear in the paper for how many in-distribution and out-distribution images (for example, as a percentage) one can decide earlier on. This percentage would be nice to show for future revisions of the manuscript.


### Minor:
The manuscript needs further polishing. For instance, the authors write, “The unnormalized logits is likely …”

Using the symbol OOD for both employed scores in eq 1 and 2 is not ideal. How about using a symbol like S with the subscript referring to the score and the superscript referring to which gate $g$ is used in the sum? It also needs to be stated that the score is per gate.
“In general, we observe that the SkipOOD variants outperform baselines like MOOD, DDU, and MSP by a considerable margin across different configurations.” By how much exactly? The results are presented in the supplementary, so it is not a helpful statement if the authors do not provide any specific measure of the “considerable margin” and specify which metric and which configurations.

# References

[1] EViT: Expediting Vision Transformers via Token Reorganizations, Youwei Liang et al. ICLR 2022 spotlight. Code and weights: https://github.com/youweiliang/evit

[2] Moreau, T., Massias, M., Gramfort, A., Ablin, P., Bannier, P. A., Charlier, B., ... & Vaiter, S. (2022). Benchopt: Reproducible, efficient and collaborative optimization benchmarks. Advances in Neural Information Processing Systems, 35, 25404-25421.

[3] Fort, S., Ren, J., & Lakshminarayanan, B. (2021). Exploring the limits of out-of-distribution detection. Advances in Neural Information Processing Systems, 34, 7068-7081.

[4] Park, J., Chai, J. C. L., Yoon, J., & Teoh, A. B. J. (2023). Understanding the feature norm for out-of-distribution detection. In Proceedings of the IEEE/CVF International Conference on Computer Vision (pp. 1557-1567).

---

> ### Author Response · Authors · 2025-01-09
> **Response to Reviewer sbH5**
>
> We thank the reviewer for their feedback.
>
> **Baselines and Benchmarks**:
>
> Thank you for the suggestion to incorporate more baselines. We will extend the paper to include ResNet-50 neural architecture and the ImageNet ID dataset to provide a more comprehensive evaluation. Currently, because of time constraints, we are training a ResNet-110 model with a Tiny ImageNet ID dataset (Size 64 X 64) to show the impact of SkipOOD with a larger input size. We will report the SkipOOD evaluation results on Texture and OpenImage-O OOD datasets by 21st January.
>
> We assert that Dynamic Neural Networks are particularly relevant in the current landscape of large neural network models, especially for real-time and edge inference applications where resources are constrained. While energy-based scores for OOD detection have been explored in the literature, our approach uniquely repurposes binary gates to serve a dual function: they not only decide which layers to skip but also enhance OOD detection. This innovative application extends beyond their traditional use, highlighting a novel contribution to the field.
>
>
> **Table 1**:
>
> For our preliminary study, we only used the SVHN dataset; however, for the evaluation, ten datasets have been used. We will make this aspect clear in the revision.
>
> **Figure 2**:
>
> We are reevaluating the effectiveness scores of MSP and will report by the 21st of January.
>
> **Undefined abbreviations of OOD scores/Motivation**:
>
> We will improve the manuscript to clarify the abbreviations and motivation.
>
> **OOD Exit Detector**:
>
> We employed a specific formulation in the experiments to obtain the OOD detection threshold lambda at a gate: use the pth percentile of OOD scores corresponding to an ID calibration dataset. The tau values are calculated for different percentiles at different gates. We expect the thresholds to be re-calibrated for each architecture. We will revise the manuscript to make this clear. Additionally, the Line plots in Figures 3 and 4 are intended to understand the effectiveness of SkipOOD when $t$ values change.
>
> **Questionable Usefulness**:
>
> The final layer is not used for all OOD inputs. Through Figure 11 in the appendix, we find the frequency of OOD inputs using different number of residual blocks. It could be noticed that a significant number of OOD inputs use less than the maximum number of blocks, which shows the effectiveness of SkipOOD in reducing latency. The significant reduction in mean latency (compared with traditional OOD detection methods) also shows that a notable number of OOD inputs do not use the final layers.

---

### Review · Reviewer_Cv9e · 2024-12-28

**Summary Of Contributions:**

This paper proposes to reduce the inference latency cost of OOD detection using an early-exiting framework. The main novelty lies in using the existing gating mechanisms in the SkipNet architecture to estimate uncertainty, rather than relying on intermediate classifer heads. The authors demonstrate that their approach is able to achieve comparable performance to baseline full inference approaches at reduced inference costs.

**Audience:**

Yes

**Claims And Evidence:**

No

**Requested Changes:**

**Critical**
1. ImageNet experiments -- there is SkipNet code for this dataset and you can use OpenOOD for evaluation. Evaluation on a single ResNet is sufficient.
2. Clarification of approach -- please present the inference process in an Algorithm box and clarify the above questions.
3. Check the MSP results and discuss why MOOD performs so much worse than the original paper. Also, add the numerical results of MOOD using the original MSDNet model for CIFAR (you can just copy from their paper).
4. Add discussion/empirical comparison to window-based early exiting for uncertainty estimation. It should be substituted into the existing inference process (although you may need to reduce the number of exits/do a bit of tuning as the original paper sets the windows heuristically). Window-based early exiting should also be applied to a cascaded ensemble of 2 ResNet 74s (the additional model should be quick to train on CIFAR) and be compared to SkipOOD.
4. Report top 1 accuracies on ID data for each model.

**Non-critical**

1. Add discussion to related work in misclassification detection and failure detection.
2. Add discussion of uncertainty score combination by normalisation via their CDFs on ID data.
3. Use more decimal places for latency results.

**Strengths And Weaknesses:**

I would like to preface by noting that due to it being the holiday season, I wrote this review whilst being rather strapped for time, and was not able to dedicate as much attention as I would have liked. I welcome the authors to correct any misunderstandings and missed/incorrect details in the review, as well as to seek any necessary clarification from me due to unclear writing in the review.

**Strengths**

- I like the idea of leveraging existing intermediate gating mechanisms to perform OOD detection. An approach like this is relatively lightweight compared to having to train additional intermediate classifier heads.
- The prose is clean and easy to follow.

**Weaknesses**
- There are no ImageNet experiments, which I view as a basic requirement for OOD detection papers in 2024.
- The details of the approach are unclear to me. What purpose do $\tau_g$ and $\tau_p$ serve in the inference process? How are $\lambda_g$ set for each gate, are they all the same? Which threshold is varied to calculate AUROC ($t,\tau,\lambda$)? Are the gates still active for performing block skipping? How are the scores at the gates and final layer combined for **SkipOOD(X,X)**? Does the inference computation cost depend on the OOD rejection threshold i.e. will more/less strict rejection thresholds mean faster/slower inference? The paper would benefit from having the inference process explicitly laid out as an Algorithm.
- The paper contains some unexpected results that don't seem to be discussed -- MOOD greatly underperforms compared to its original paper, and the MSP score seems to be more uncertain on ID data for some reason.
- Missing comparison to window-based early exiting for uncertainty estimation [1] -- this is an exit criterion that can be applied to any model with intermediate uncertainty estimates, that relies on the assumption that samples that lie within a
*window* of uncertainty around the rejection threshold will most likely be improved by later stages/exits. How does this exiting criterion behave compared to the proposed patience-based method? How does a window-based cascaded ensemble compare to SkipOOD?
- Missing discussion of related work in misclassification detection [2,3], and failure detection [4,5] (rejecting both ID errors *and* OOD).
- The OOD detection performance results presented are difficult to interpret -- Fig. 2 would be better replaced with a table of numerical values.
- SkipOOD seems to require an OOD validation set to tune hyperparameters (of which there are a quite a few which all seem to affect the OOD detection performance).
- Different uncertainty scores can be combined by using their cumulative density functions on ID data [6]. This should be added to the discussion at the end of page 4.

[1] Xia & Bouganis, Window-Based Early-Exit Cascades for Uncertainty Estimation: When Deep Ensembles are More Efficient than Single Models, ICCV 2023

[2] Hendrycks & Gimpel, A Baseline for Detecting Misclassified and Out-of-Distribution Examples in Neural Networks, ICLR 2017

[3] Geifman & El-Yaniv, Selective Classification for Deep Neural Networks, NeurIPS 2017

[4] Xia & Bouganis, Augmenting Softmax Information for Selective Classification with Out-of-Distribution Data, ACCV 2022

[5] Jaeger et al., A Call to Reflect on Evaluation Practices for Failure Detection in Image Classification, ICLR 2023

[6]  Xue et al., Enhancing the Power of OOD Detection via Sample-Aware Model Selection, CVPR 2024

---

> ### Author Response · Authors · 2025-01-09
> **Response to Reviewer Cv9e**
>
> We thank the reviewer for the comments.
>
> **Inference Algorithm**:
>  Below, we provide a detailed description of the inference algorithm. In brief, features extracted from each gate are used to generate OOD scores, which are then compared against a gate-specific threshold ($\lambda_g$) to determine if the input is OOD at that gate. The computation is terminated if the total number of gates identifying the input as OOD exceeds another threshold $t$. Although there are various methods to determine $\lambda_g$, in our experiments, we calibrate it using an ID dataset. Specifically, we set $\lambda_g$ to the percentile value of the OOD scores at a gate corresponding to the calibration set. The percentile value $\tau_g$ and count threshold $t$ are varied when calculating AUROC. Block skipping is activated during inference, and the computational cost depends on the thresholds, representing a trade-off between OOD accuracy and early termination. In the revised draft, we will add an algorithm block.
>
> **Limited effectiveness of MSP/MOOD Scores**:
> MOOD uses the bit-length of the encoded input to assess input complexity and determine the appropriate output exit. The effectiveness of these exits is influenced by both the architecture and their positions within the network. In our investigation of EENet early exit ResNet models, we discovered that not all exits are well-suited for detecting OOD data. We are reevaluating the effectiveness scores of MSP and will report by the 21st of January.
>
> **Additional Discussions, References and Clarity**:
> We will include additional discussions for window-based early exiting and cumulative density functions.  We will also incorporate the specified references and improve Figure 2 for clarity.
>
> **ImageNet Experiments**
> We will extend the paper to include ResNet-50 neural architecture and the ImageNet ID dataset to provide a more comprehensive evaluation. Currently, because of time constraints, we are training a ResNet-110 model with a Tiny ImageNet ID dataset (Size 64 X 64) to show the impact of SkipOOD with a larger input size. We will report the SkipOOD evaluation results on Texture and OpenImage-O OOD datasets by 21st January.

---

> > ### Comment · Reviewer_Cv9e · 2025-01-13
> >
> > Thanks for getting back to me.
> > I'm happy to give the authors extra time to run experiments to improve the paper if the action editor permits.
> > The ImageNet experiments are a hard requirement for me.

---

### Author Response · Authors · 2025-01-21
**Rebuttal for Additional Experiments**

We thank Reviewer Cv9e for their feedback. As noted, conducting experiments on ImageNet is a critical requirement. We kindly request additional time to perform these experiments or the opportunity to submit a revised version of the paper that includes these results.

Regarding the MSP evaluation, we retrained two architectures—ResNet110 and ResNet74—with CIFAR-10 and CIFAR-100 in-distribution (ID) data. For the baseline experiments, we utilized the implementation provided in the NECO framework [1] (https://gitlab.com/drti/neco). The updated results are presented in the table below:

| Model           | SkipOOD (D,E) | ML   | MSP  | Energy | KL   | MOOD |
|------------------|---------------|-------|-------|--------|-------|-------|
| RN110-CIFAR10   | 0.84          | 0.88  | 0.85  | 0.88   | 0.77  | 0.50  |
| RN110-CIFAR100  | 0.79          | 0.81  | 0.73  | 0.81   | 0.72  | 0.63  |
| RN74-CIFAR10    | 0.87          | 0.88  | 0.87  | 0.88   | 0.79  | 0.51  |
| RN74-CIFAR100   | 0.79          | 0.82  | 0.74  | 0.82   | 0.76  | 0.51  |


Upon reevaluation, we identified an error in the intermediate outputs used for the MSP calculations in the initial submission. These errors have been corrected in the table above. The evaluation results for SkipOOD and the other baseline methods are consistent with the values reported in the paper, though not identical. SkipOOD demonstrates competitive out-of-distribution (OOD) detection AUROC compared to baseline methods that utilize the full network.

Furthermore, our reevaluation of mean latency aligns with the results presented in Table 1 of the paper. The primary reason for this is that most baseline methods utilize the full network, resulting in no change in mean latency for those approaches. For SkipOOD (D,E) and MOOD we also find the same mean latency as reported in Table 1.  This confirms that SkipOOD effectively detects OOD samples while utilizing only 50\% of the full network latency.

[1] Ammar, Mouin Ben, et al. "NECO: NEural Collapse Based Out-of-distribution detection." arXiv preprint arXiv:2310.06823 (2023).

---

### Decision · Action_Editor_kXvQ · 2025-02-18

**Recommendation:** Reject

**Comment:**

The authors did not provide convincing enough empirical evidence that SkipOOD is effective compared to prior work. To improve the draft for a resubmission, the authors should follow reviewer suggestions, especially by addressing the above two major weaknesses.

**Audience:**

Yes. OOD detection is a well-studied field within ML and the paper's goal is to advance state-of-the-art OOD detection performance using a new technique.

**Claims And Evidence:**

While reviewers generally find the method itself novel and intuitive, current evaluation is missing critical supporting evidence for its effectiveness. Some of the major weaknesses include:
- Lack of evaluation on large-scale datasets such as ImageNet and missing baselines (Reviewers Cv9e and sbH5).
- Inconsistent result compared to prior work (Reviewers Cv9e, sbH5 and a7he).

The authors did not sufficiently address these concerns in their rebuttal. As a result, reviewers leaned towards rejection as the paper currently does not provide enough convincing evidence.

**Resubmission Of Major Revision:**

The authors may consider submitting a major revision at a later time.